# Haisu: Hierarchically supervised nonlinear dimensionality reduction

**Kevin Christopher VanHorn**[ID][1]*, **Murat Can Çobanoğlu**[ID][1]*

Lyda Hill Department of Bioinformatics, University of Texas Southwestern Medical Center, Dallas, Texas, United States of America

* kevin.vanhorn@utsouthwestern.edu (KV); muratcan.cobanoglu@utsouthwestern.edu (MC)

## Abstract

We propose a novel strategy for incorporating hierarchical supervised label information into nonlinear dimensionality reduction techniques. Specifically, we extend t-SNE, UMAP, and PHATE to include known or predicted class labels and demonstrate the efficacy of our approach on multiple single-cell RNA sequencing datasets. Our approach, "Haisu," is applicable across domains and methods of nonlinear dimensionality reduction. In general, the mathematical effect of Haisu can be summarized as a variable perturbation of the high dimensional space in which the original data is observed. We thereby preserve the core characteristics of the visualization method and only change the manifold to respect known or assumed class labels when provided. Our strategy is designed to aid in the discovery and understanding of underlying patterns in a dataset that is heavily influenced by parent-child relationships. We show that using our approach can also help in semi-supervised settings where labels are known for only some datapoints (for instance when only a fraction of the cells are labeled). In summary, Haisu extends existing popular visualization methods to enable a user to incorporate labels known *a priori* into a visualization, including their hierarchical relationships as defined by a user input graph.

**Data Availability Statement:** The authors declare that the data supporting the findings of this study are available within the paper and its supplementary information files. Code is available at https://github.com/Cobanoglu-Lab/Haisu.

## Author summary

Information collection and representation is critical in modern applications as advances in technology alleviate storage and computational processing constraints. Large quantities of information often manifest as high-dimensional data that is too complex to understand. One can employ non-linear dimensionality reduction so that the data can be viewed in two or three dimensions. Often, however, this visualization process can miss key relationships in the data that are already well-understood or known beforehand. With our algorithm, Haisu, we aim to guide these dimensionality reduction approaches by integrating a hierarchy graph based on categories in the original high-dimensional information. Our method can be customized so that a user can explore a dataset's visualization with varying degrees of influence of the input hierarchy. This approach is especially valuable in the biological domain where important cell-to-cell relationships can easily be obfuscated by dimensions introduced by irrelevant genes. Thus, we aim to introduce a user-guided

**Funding:** The authors received no specific funding for this work.

**Competing interests:** The authors have declared that no competing interests exist.

modification to visualization techniques that can integrate an input hierarchy to better understand data in the context of known relationships.

This is a *PLOS Computational Biology* Software paper.

## Introduction

Dimensionality reduction (DR) involves the transformation of features or random variables of a given dataset to a lower dimension. DR is a crucial component in modern analysis techniques and is widely applicable due to the rapid growth in data available to businesses, scientists, and public administration [1]. This data can be comprised of text-based or multimedia content from entertainment, research, and business sectors. The role of DR for feature selection and extraction and data preprocessing is especially pertinent when analyzing immense volumes of data with methods such as machine learning [2].

We focus on the use of dimensionality reduction for visualization of biomedical data. When high-dimensional data is difficult to summarize, DR is often effective for transforming the data to two or three dimensions. This can be achieved through linear and nonlinear approaches. Nonlinear dimensionality reduction (NLDR) is preferable for capturing the local and global structure of the data, where linear DR tends to be faster and more effective solely for global patterns [2]. Our method, Haisu, introduces a general-purpose NLDR extension based on three popular visualization methods that incorporates an input hierarchy. Haisu can be manually weighted with a strength factor to better represent patterns in the data at any degree of influence. We implement this technique for both known and predicted class-based hierarchies with the option to influence the strength of a predicted class' structural effect based on its likelihood.

Many nonlinear techniques exist for dimensionality reduction and are mainly unsupervised. Examples of such methods include Sammon mapping [3], Curvilinear Components Analysis (CCA) [4], Stochastic Neighbor Embedding (SNE) [5], Isomap [6], Maximum Variance Unfolding (MVU) [7], Locally Linear Embedding (LLE) [8], and Laplacian Eigenmaps [9]. These are often effective with artificial data but struggle to maintain both local and global structure on real-world data [10]. For this reason, we primarily target modern nonlinear dimensionality reduction methods. For demonstrational purposes, we show the effect of Haisu on Principal Component Analysis (PCA) as well, a widely used unsupervised linear technique.

To our knowledge, no modification has been introduced to enable a general-purpose NLDR technique to be influenced by a user-defined hierarchy. Auxiliary kernel-based semi-supervised techniques can be effective at preserving the local and global structure of data and have recently shown effective in sc-RNA sequencing data [11]. Haisu is independent of such pre-processing method and can even be used in a complementary manner, preserving their effects while introducing a supervised hierarchy-based component. Other supervised and semi-supervised DR approaches exist but are not readily applicable to general-purpose visualization [12,13]. Finally, separate techniques that integrate hierarchical information such as Hierarchical Manifold Learning are less concerned with visual analysis in lower dimensions and thus not as relevant to our work [14]. We aim to provide a context-independent approach that incorporates a tunable supervised hierarchical influence across stochastic embedding approaches. We argue that if a hierarchy for a dataset is known *a priori*, it can provide benefit when integrated into the analysis. For example, such

an approach can help to alleviate the case where important and explicit relationships in the data are ignored due to low expression.

We want to describe the problem we are solving with one concrete example. Assume a scientist has a single-cell RNA sequencing (scRNA-seq) dataset of peripheral blood mononuclear cells (PBMC). It is a well-established fact, through decades of research and countless articles, that there are T cells, B cells, monocytes etc. Furthermore, it is also well established that said T cells have subtypes such as $CD4^+$, $CD8^+$ and so on. It is also clear that these are subtypes of T cells and not B cells or the others. Currently, despite the extensive knowledge about these hierarchical relationships, the popular visualization techniques t-SNE, UMAP and PHATE have no mechanism for imparting that information into the visualization. We developed Haisu to solve this problem.

In an ideal scenario, the NLDR methods should pick up on any hierarchy unaided. However, in the practical world, there are many technical problems with data collection in every possible experiment. Examples include pipetting errors, dropout or low read counts in scRNA-seq, or poor antibodies in CyTOF. When any problem in the data collection causes the NLDR method to fail to pick up on the latent hierarchy, the relationships among the various cell types still hold true—$CD4^+$, $CD8^+$ T cells are still T cell subsets, and different from B cells—they are simply erroneously visualized. With Haisu, the user can directly correct such errors.

To explore the benefits of integrating a supervised hierarchical approach for DR, we primarily focus on scRNA-seq datasets in this manuscript. Nonlinear dimensionality reduction is widely applied to sc-RNA sequencing, and many such datasets are influenced by an interpretable cell-based hierarchy known prior to experimentation. Because clusters identified via NLDR are frequently integrated into unsupervised cell-type prediction models, we suggest that Haisu is particularly useful in this domain [15–17]. We highlight three state-of-the-art NLDR approaches: t-SNE, UMAP, and PHATE.

Formulated by Maaten and Hinton, t-SNE is currently the premier dimensionality reduction method used for high-dimensional data visualization [18]. This technique is a variation of Stochastic Neighbor Embedding, or SNE, which converts the Euclidian distance between high-dimensional data points into conditional probabilities that represent similarities. T-SNE's variation adds a Student's t-distribution to SNE which solves the crowding problem that standard SNE suffers from. Uniform Manifold Approximation and Projection (UMAP) is competitive to t-SNE and aims to better preserve the global structure of data with better scalability. Finally, potential of heat diffusion for affinity-based transition embedding (PHATE) is a domain-specific visualization strategy designed for biological datasets [19]. PHATE aims to improve denoising and provide biological insight across datasets and data accession methods.

## Design and implementation

**Hierarchically supervised NLDR.**   Given a high dimensional input $\mathcal{X} = \{x_1, x_2, \ldots, x_N\}$, the asymmetric similarity $s_{j|i}$ between $x_i, x_j$ is commonly determined in-part by employing a distance function $d(x_i, x_j)$. Furthermore, given an undirected (not necessarily connected) input hierarchy graph $G = (V_G, E_G)$ with labels assigned to each node, we process input high-dimensional points $x_i, x_j$ with labels $i', j'$ respectively.

Our algorithm introduces $\theta_{ij}$ as a direct modification to the distance function based on the input label hierarchy where $0 \leq \theta_{ij} < 1$. As applied to an existing nonlinear dimensionality

technique, Haisu produces a modified pairwise distance function:

$$d'(x_i, x_j) = \theta_{ij} \cdot d(x_i, x_j)$$

To calculate $\theta_{ij}$, we find the shortest path distance of two undirected nodes $\overrightarrow{j'i'}$ in $G$. Define *maxdist* as the maximum length shortest path in $G$; in other words, *maxdist* is defined as *max* ($S$) where $S$ is the set of possible shortest paths in $G$. Finally, let *str*$\in[0,1)$ denote a parameter to modulate the "strength" of the hierarchical distancing factor $\theta_{ij}$ as applied to the pairwise-distance function $d(x_i, x_j)$. At lower values of *str*, the distance $d(x_i, x_j)$ is preserved, and at higher values, distances are penalized based on the corresponding hierarchy graph. The resulting function is as follows:

$$\theta_{ij} = (1 - str) + \frac{str \cdot \overrightarrow{j'i'}}{maxdist}$$

Using this formulation, a user can choose to input the adjacency matrix of the input graph $G$ and the strength of the hierarchical distancing factor. If a label does not exist for $x_i$ or labels $i'$, $j'$ are in different connected components, then $\overrightarrow{j'i'} = maxdist$ and $\theta_{ij} = 1$. This ensures that regardless of the hierarchical distancing factor, the original pairwise distance for that node is maintained. In our implementation, graphs can be specified for columns or rows of the input dataset. In the case that there is a known hierarchy for the dataset, but labels are unknown, we specify a feature-based method that weights the strength of a predicted class' structural impact based on its likelihood. Given the probabilities $p_i$ and $p_j$ of labels $i'$ and $j'$ for high-dimensional points $x_i$, $x_j$, we define $m_{ij} = min(p_i, p_j)$. We then weight the distancing factor $\theta$ with $m_{ij}$:

$$\theta_{ij} = \left(1 - m_{ij} \cdot str\right) + \frac{m_{ij} \cdot str(\overrightarrow{j'i'} - 1)}{maxdist - 1}$$

Our reference software is implemented in Python 3. We provide a modified pairwise distance matrix to any NLDR approach that accepts a 'precomputed' metric. The input parameters for our implementation include an adjacency list defining the class hierarchy, the strength factor *str*, a standard pairwise distance function from the scikit-learn library [20], a list of class labels, and a list of class probabilities when relevant. We also include several parameters to drive the effect of the hierarchy graph $G$: *disconnected-distance*, *self-distance*, and an array of unique labels *avoid-self*. Finally, users can specify the explicit weights of edges prior to normalization of the "path-pairwise" matrix.

**Graph construction for Haisu.** We construct a "path-pairwise" matrix $H = (h_{ij})$, where $h_{ij} = \theta_{ij} \cdot d_{ij}$, that modifies the raw distance matrix $D = (d_{ij})$, where $d_{ij} = d(x_i, x_j)$. $H$ is normalized between [0,1] which has implications on the choice of hierarchy graph regarding inter-class distance, intra-class distance, and the effect of explicit or implicit edge weights on the resulting matrix $H$.

To determine intra-class distances, one can provide a set of labels with the parameter *avoid-self* $\subseteq V_G$ which specifies $\theta_{ij} = $ *self-distance* for vertices in $G$, given $i' = j'$. This modification is applied after normalization of $H$ for classes in *avoid-self*, s.t $0 \leq$ *self-distance* $\leq 1$. By default, this parameter is 0 which will exaggerate the hierarchical effect in the embedding for high-dimensional points with the same label (Fig 1B). Beyond the use of *str*, one can reduce the effect of Haisu on an NLDR technique by increasing *self-distance* above 0 so that $\forall x_i$, $x_j \in X \rightarrow \theta_{ij} > 0$ within a connected component of $G$. Thus, in situations where it is more desirable to preserve characteristics of the NLDR technique or reduce clustering, one should consider increasing *self-distance* and adding respective classes to *avoid-self*. In Fig 1 we demonstrate the effect of modulating this value on a set of random points.

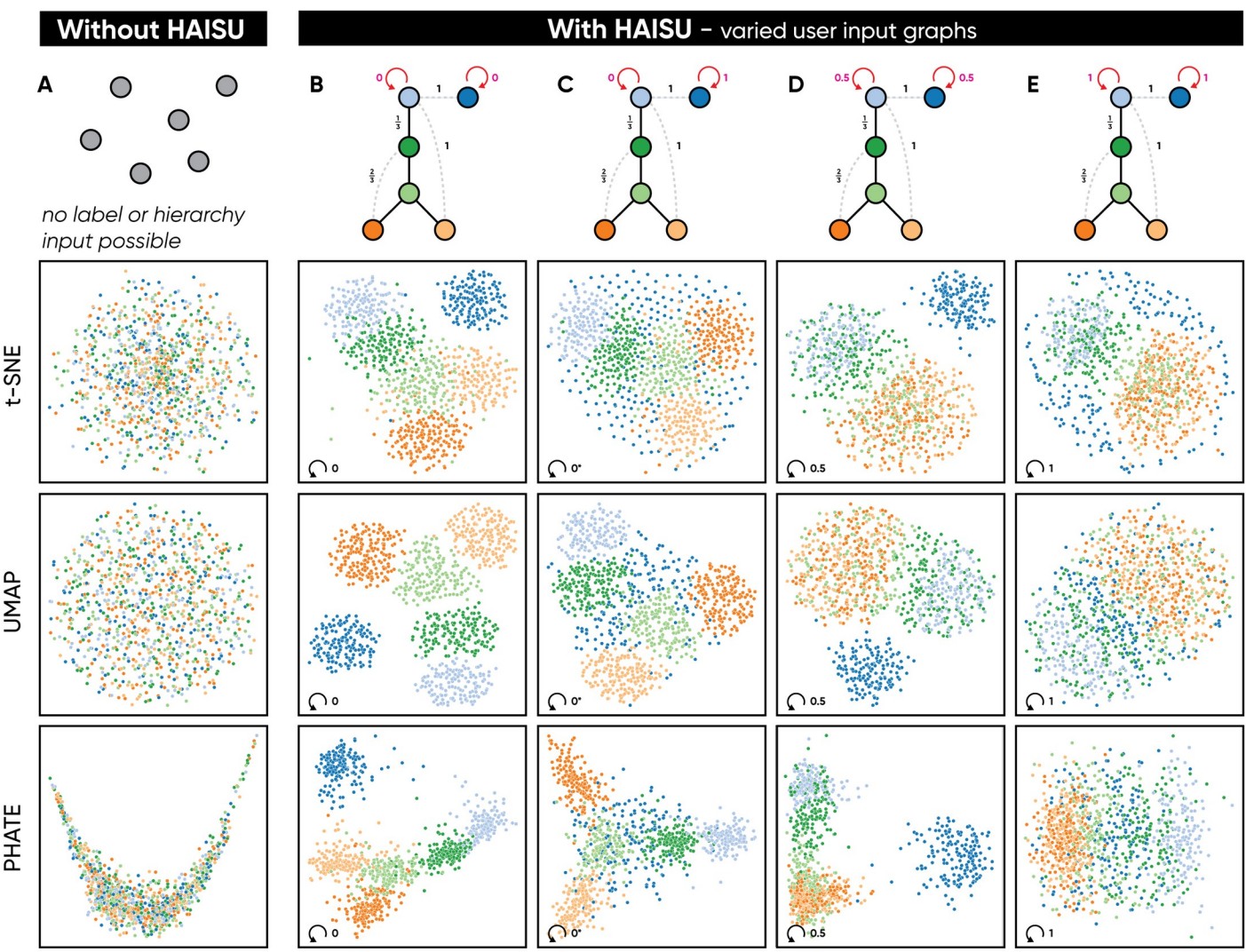

**Fig 1. Users can control the effect of an input hierarchy on the resulting embedding.** We demonstrate the effect of Haisu as applied to t-SNE, UMAP, and PHATE with an input graph on a set of random points. In (A) we display the unmodified embedding of each NLDR method without HAISU or any hierarchical prior. In (B), we demonstrate the default mode of Haisu where self-distance = 0; at higher str values this results in a stronger hierarchical effect on the original embedding. In (C) self-distance = 1 for the disconnected class (blue), which is penalized for clustering near itself and spreads back toward classes each point is most similar to. In (D) self-distance = 0.5 which still applies the hierarchy but encourages more inter-class interaction than (B). Finally, in (E) self-distance = 1 for all classes, which penalizes intra-class clustering, resulting in a much looser representation of the hierarchy compared with (B).

Inter-class distances are determined by edge weights and the size of $G$, as applied via the path-pairwise matrix $H$. To better convey this effect, we examine the hierarchical distance factor $\theta_{ij}$ at $str = 1$ where $0 \leq \theta_{ij} \leq 1$. The bounds of this range are dependent on the distance $\overset{\frown}{j'i'}$ so that if $i \neq j$:

$$\frac{\overset{\frown}{j'i'}}{maxdist} \leq \theta_{ij} \leq 1$$

The effect of the inter-class shortest path distance $\overset{\frown}{j'i'}$ of labels $j'$ and $i'$ is contingent upon the normalization factor *maxdist*. In the implicit case where no edge weights are provided (Figs 2–4), it is important to note that $\theta_{ij}$ will be inversely affected by *maxdist* which tends to increase with the size of the graph $G$. As a result, high-dimensional pairwise distances between

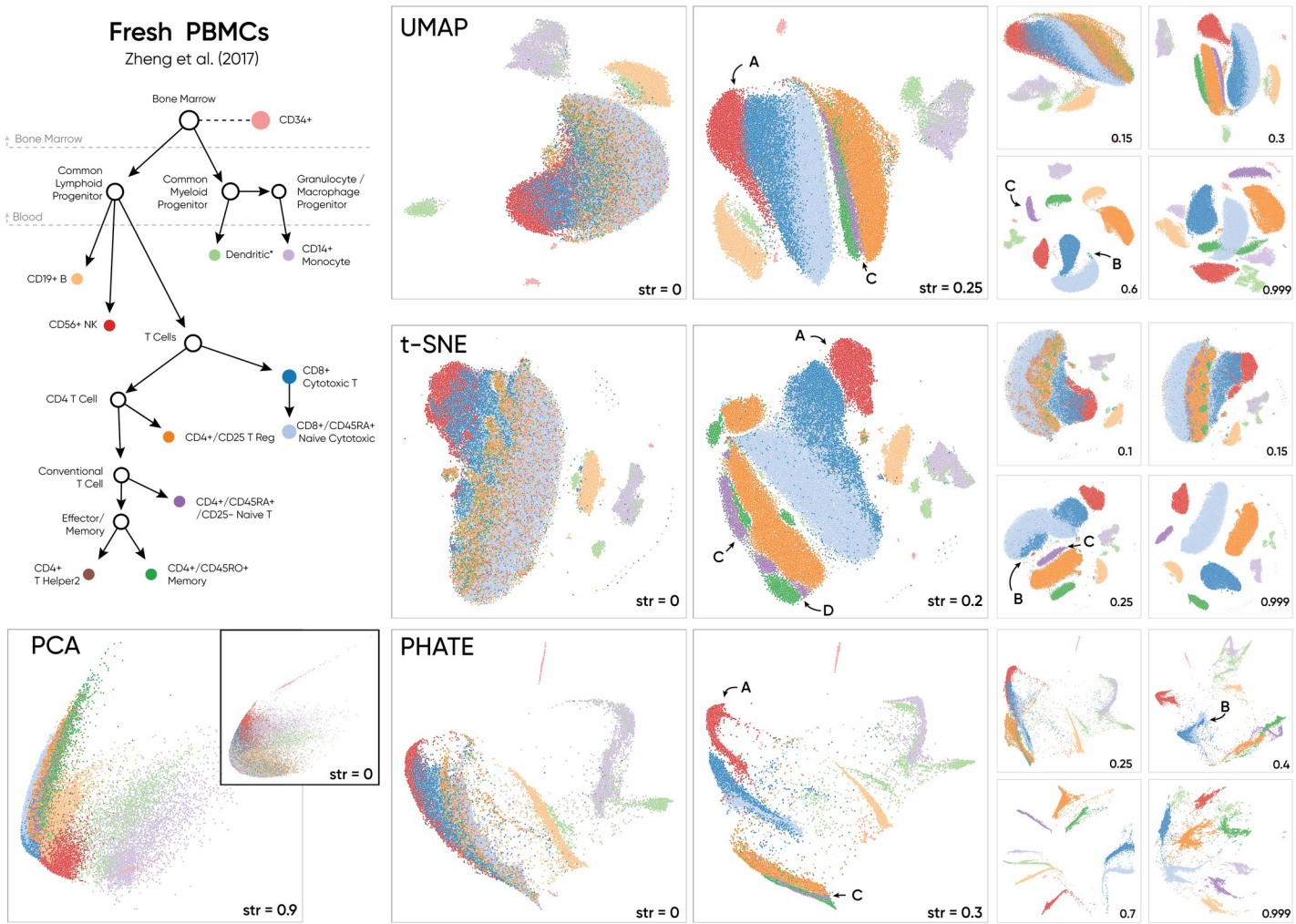

**Fig 2. For this dataset, Haisu applies an input hierarchy based on cell function and lineage to guide the identification of sub-clusters.** We display the effect of our method on popular nonlinear DR approaches and PCA at multiple 'strength' values (str), a tunable factor between 0 and (up to) 1 to control the strength of our hierarchical distancing function. Compared to raw NLDR (str = 0), Haisu reveals sub-clusters of T cells and better expresses the subtle relationship between datapoints in each method.

$x_i$ and $x_j$ will tend to be closer as the size of their corresponding hierarchy graph increases when $str > 0$. If this effect is not desirable, one can influence inter-class distances $\overrightarrow{j'i'}$ per edge by adding null/dummy nodes that do not correspond to a label to artificially weight connections. Alternatively, one can specify edge weights prior to normalization of $H$. For example, in a graph that contains both general and specific labels, one may not want all nodes to be distanced equally in the hierarchy graph. A "general" label here corresponds to an abstract class that represents multiple sub-classes not in the graph $G$.

Finally, the parameter *disconnected-distance* $\geq 0$ determines $\theta_{ij}$ when $j'$ and $i'$ exist in disconnected components of $G$. This value is applied prior to a min-max normalization of $H$ between 0 and 1 and is not affected by *maxdist*. By default, this value is 1 but can be increased to greater exaggerate distances between connected components. As a result, when *disconnected-distance* is greater than 1, the effect of inter-class distances within connected components will be lessened in $H$ due to the accompanying min-max normalization.

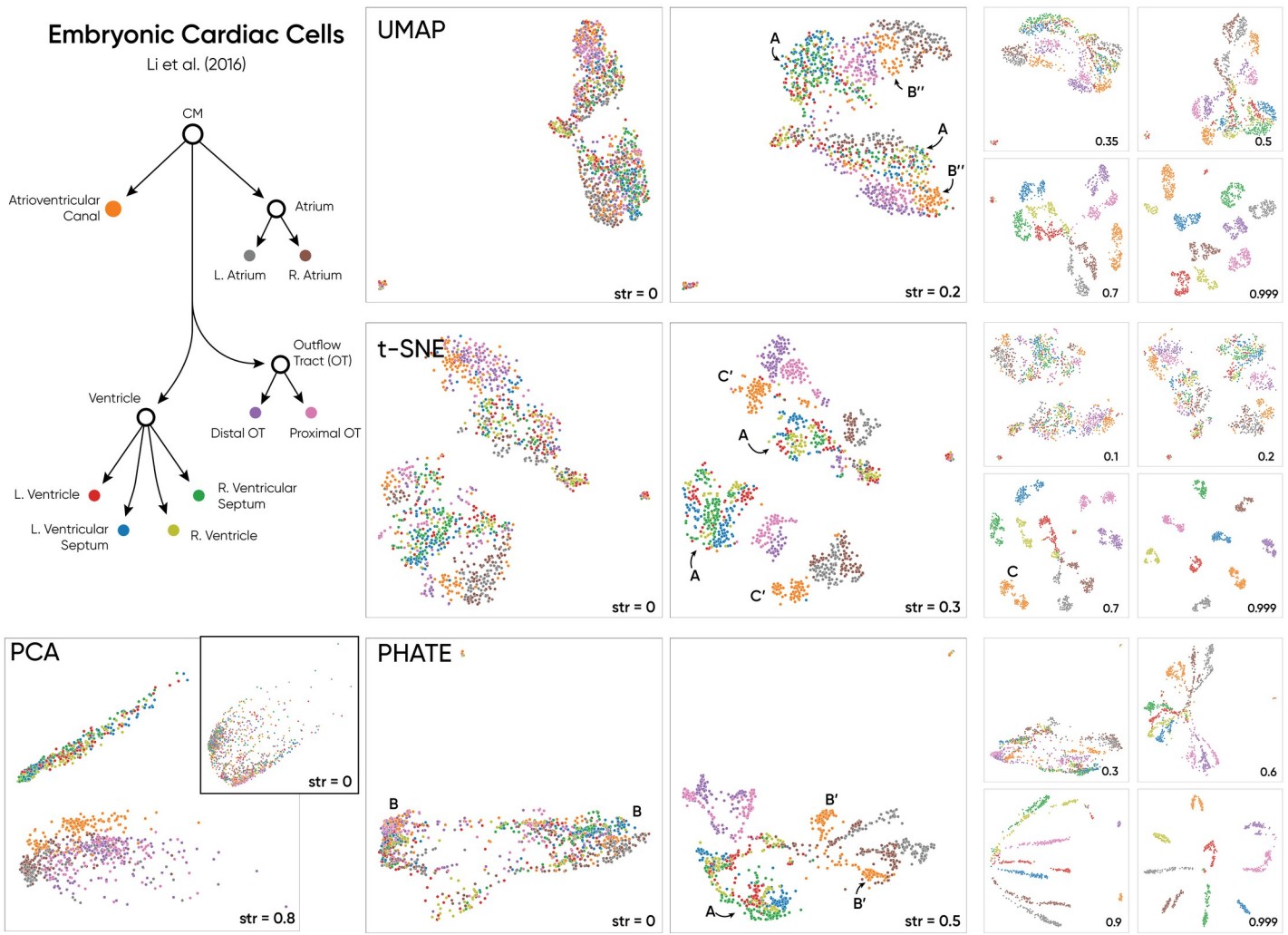

**Fig 3. Haisu applied to anatomical embryonic cardiac cell subpopulations via a proximity-based hierarchy.** The raw embeddings of each method indicate two primary clusters with cell label assignments that are spaced out within each cluster. Haisu helps to add clarity to the embedding in a manner true to the known external hierarchy. Labels are assumed to be 100% accurate as they are location-based, but anatomic regions can have similar transcriptomic profiles. Thus, Haisu in this context, factors in gene expression and location when determining a lower dimensional embedding at an appropriate strength.

**The effect of Haisu on t-SNE, UMAP, & PHATE.** Haisu's hierarchical distancing factor $\theta$ modifies high-dimensional distances to influence downstream analysis of the input data while taking the input hierarchy into account. We describe the repercussions of this modification to t-SNE, UMAP, and PHATE where $s_{i|i} = 0$ for a user-specified distance function $d(x_i, x_j)$.

To examine the effect of Haisu on the integrity of a resulting embedding for a general NLDR technique, we first describe $\theta_{ij}$ when $i' \neq j'$. For the base case, $\theta_{ij} = 1$, the raw NLDR pairwise similarities are preserved. When $0 < \theta_{ij} < 1$, Haisu will have a downstream effect on the final embedding by reducing the pairwise distances between $x_j$ and $x_i$ and increasing pairwise similarity. This affects the joint probability distribution $\hat{p}_{ij}$ by strictly reducing $\hat{p}_{ij}$ relative to $p_{ij}$. To minimize potential crowding in the embedding and better represent the original NLDR method at higher values of *str*, Haisu will benefit from a lower user-provided *perplexity*, *n_neighbors*, or *knn* value which estimate or determine the number of nearest neighbors during the calculation of asymmetric similarities $\hat{p}_{j|i}$ for t-SNE, UMAP, and PHATE respectively.

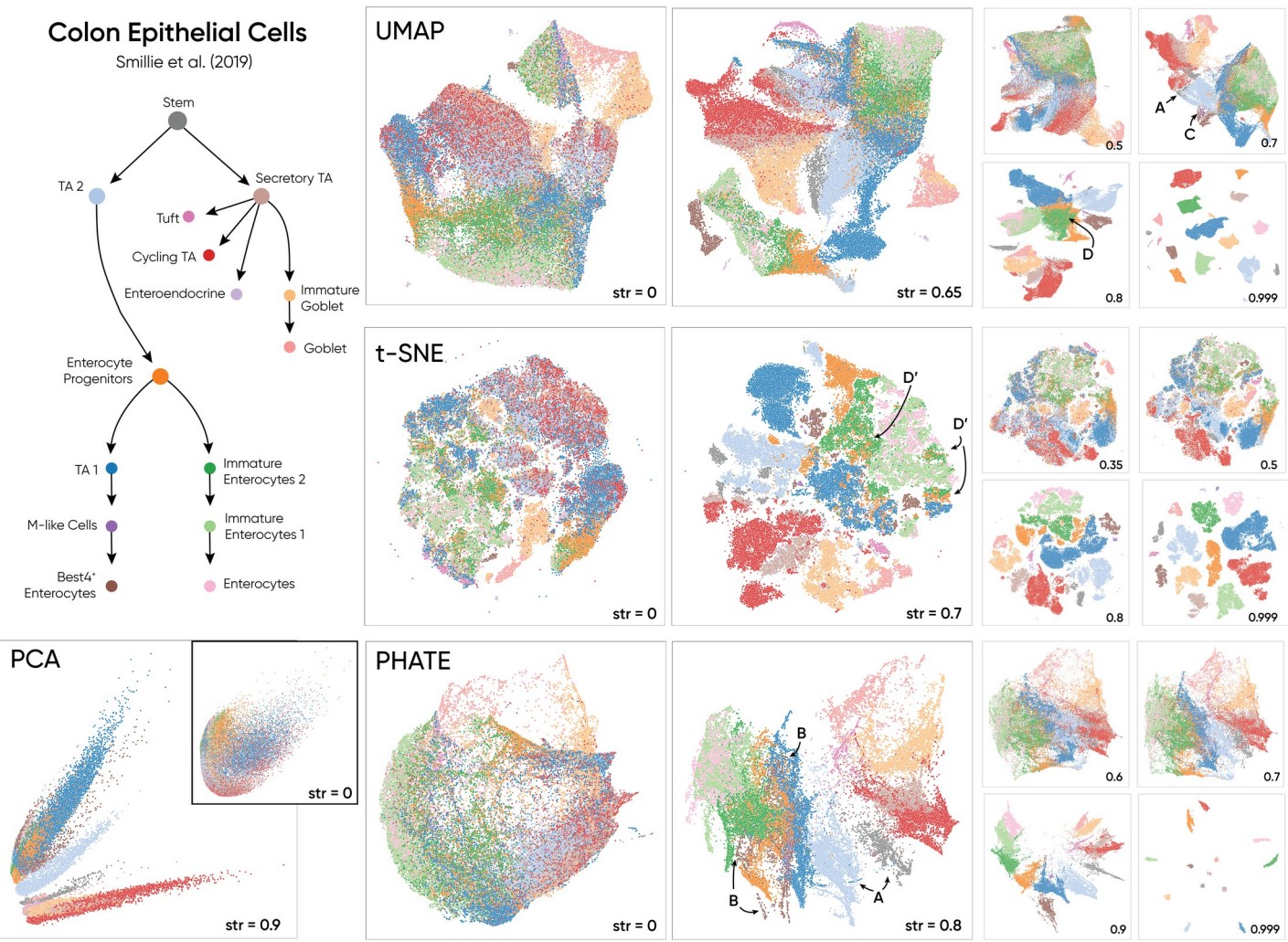

**Fig 4. We illustrate the effect of Haisu within the context of an epithelial differentiation hierarchy in the context of healthy and ulcerative colitis patients.** In this dataset, strength factors up to 0.8 uphold appearance of the raw embedding. Thus, with sufficient confidence in cell type labels, Haisu preserves the structure of the NLDR method while also allowing a simpler examination of more subtle inter-cluster relationships.

The minimum boundary of the hierarchical distancing factor $\theta_{min} = Min(H)$ is driven by the minimum edge distance $\overrightarrow{j'i'}_{min}$ or the user's choice of parameter *self-distance*, which specifies $\theta_{ij}$ when $i' = j'$. In this regard, within the same connected component of $G$, Haisu will have one of the following cases if *str* = 1:

$$\textbf{Case 1}: \ \theta_{min} = 0$$

$$\textbf{Case 2}: \ \theta_{min} = self-distance$$

$$\textbf{\textit{Case 3}}: \ \theta_{min} = \frac{\overrightarrow{j'i'}_{min}}{maxdist}$$

Case 1 is the implicit configuration of Haisu in the presence of a uniform-weighted graph, or given *self-distance* is not specified. Case 2 is determined by the user-provided parameter for intra-class distances when $0 < self\text{-}distance \leq 1$ (Fig 1) given that any provided weights do not

exceed *self-distance*. Case 3 occurs at higher values of *self-distance*, when normalized graph distances are less than intra-class distances. At high values of str and/or low values of *self-distance*, Haisu can destructively modify the integrity of the NLDR so that intra-class distances are drastically different after application of *H*. For all the specified NLDR methods in this section, asymmetric similarity calculation is critical to downstream analysis and case 1 will have the following effect: $\hat{s}_{j|i} = 1 | \theta_{ij} = 0$. This will also affect the cost function $\hat{C}_{ij} = 0 | i\prime = j\prime$ so that neither low nor high-dimensional coordinates are employed between points of the same class, hence our limitation of *str* < 1 (Figs 2–5).

T-SNE extends the pairwise similarity definition $s_{j|i}$ from SNE to a conditional probability $p_{j|i}$ where $\sigma_i^2$ is the variance of a Gaussian centered at $x_i$. T-SNE then determines joint probabilities $p_{ij}$ in high-dimensional space as symmetrized conditional probabilities *s.t.* $p_{ij} = p_{ji}$. With Haisu, the proportionality of $\theta_{ij}$ to $s_{j|i}$ is inversely exponential because $\theta$ is a fractional modifier bound between 0 and 1. Thus as $\theta$ decreases with closer relative distances in the hierarchy graph, $d(x_i, x_j)$ will decrease in the kernel, resulting in a higher modified similarity $\hat{s}_{j|i}$. With the introduction of the hierarchical distance factor $\theta_{ij}$, we also include the resulting conditional and joint probabilities $\hat{p}_{j|i}$, and $\hat{p}_{ij}$ respectively. We note that t-SNE [18] specifies $d(x_i, x_j)^2$, so in our implementation, scipy.spatial.distance functions [21] are supplied the default argument "squared = True," however this is not strictly enforced.

$$s_{j|i} = \exp(-d(x_i, x_j)^2 / 2\sigma_i^2)$$

$$\hat{s}_{j|i} = (s_{j|i})^{\theta_{ij}} = \exp(-\theta_{ij} \cdot d(x_i, x_j)^2 / 2\sigma_i^2)$$

$$\hat{p}_{j|i} = \frac{\hat{s}_{j|i}}{\sum_{k \neq i} \hat{s}_{k|i}} = \frac{(s_{j|i})^{\theta_{ij}}}{\sum_{k \neq i} (s_{k|i})^{\theta_{ik}}}$$

$$\hat{p}_{ij} = \frac{\hat{p}_{j|i} + \hat{p}_{i|j}}{2N}$$

T-SNE aims to minimize the separation between the data points in the original high-dimensional space and the low-dimensional projection space. As such, it makes no topological guarantees except for the implication that points close in the high-dimensional space will likewise be close to each other in the projection. Thus, the neighborhood matrix is of fundamental importance since the construction of the final embedding is focused on preserving neighborhoods in the low-dimensional projection. Using Haisu, we perturb the high-dimensional space to impart path distances in the hierarchical label graph onto the neighborhood relationships by reducing distances in high-dimensional space. The higher the global 'strength' value, local edge weight, and label probabilities assigned to pairs $x_i, x_j$, the stronger this effect. Datapoints with closer labels in the hierarchy are prioritized during the minimization of Kullback-Leibler divergence due to the reduced pairwise similarity $\hat{s}_{j|i}$ and thereby have a higher propensity to be identified as neighbors. Consequently t-SNE is modified by Haisu in such a way that the low dimensional projection reflects the input hierarchy.

Pairwise similarities in UMAP are calculated as smoothed nearest neighbor distances where $n_i$ is the distance to the nearest neighbor of $i$, and $\hat{\sigma}_i$ is the Haisu-modified normalizing factor. Without further normalization, UMAP then symmetrizes $s_{j|i}$ with a probabilistic fuzzy set

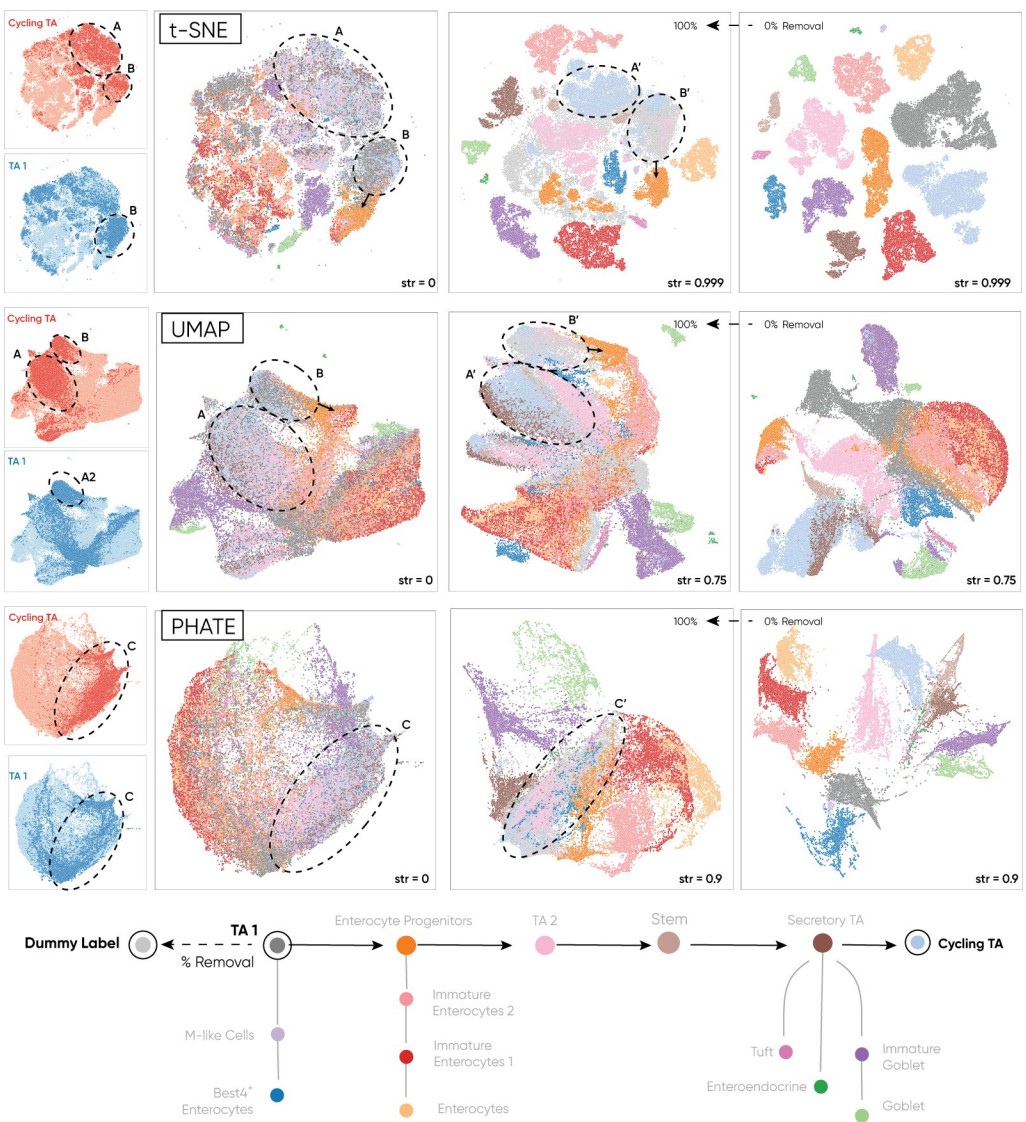

**Fig 5. Haisu does not compromise the embedding of cells that do not have a label in the input graph.** We depict 0% and 100% replacement of the TA-1 label with a 'dummy' label that is not present in the hierarchy across t-SNE, UMAP, and PHATE. Even at high strength values (str) of the hierarchical distancing factor, Haisu maintains relationships in the embedding circled in the figure. Notably, TA 1 cells remain close to Cycling TA cells across the embeddings at 100% removal despite their distance in the hierarchy graph. Thus, we do not comprise the integrity of each NLDR method, allowing for the observation of unknown classes in the context of a strongly influential, known hierarchy.

union to obtain $p_{ij}$.

$$s_{j|i} = \exp((-d(x_i, x_j) - n_i)/\sigma_i)$$

$$\hat{s}_{j|i} = \exp((-\theta_{ij} \cdot d(x_i, x_j) - \theta_{ij} \cdot n_i)/\hat{\sigma}_i)$$

$$\hat{p}_{ij} = (\hat{s}_{j|i} + \hat{s}_{i|j}) - \hat{s}_{j|i}\hat{s}_{i|j}$$

To compute the standard normalizing factor $\sigma_i$, UMAP performs a binary search for $\sigma_i$ s.t. $\sum_{i=1}^{k} \exp(-(K_i - p)/\sigma) = \log_2(k)$. This optimization to $\log_2(k)$ is an empirical measurement

by UMAP to smooth the k-nearest distances $K$. Supplying a precomputed distance matrix modified by Haisu affects the calculation of the normalizing factor $\sigma_i$ via the distance to the nearest neighbors $K$ which affects $p = K_1$. Thus, via $\hat{\sigma}_i$, UMAP takes the hierarchical matrix $\theta$ into account when smoothing the $k$ nearest neighbor distances for the pairwise similarity $\hat{s}_{j|i}$. This is necessary to fix the cardinality of the fuzzy set of 1-simplices to a fixed value influenced by $\theta$. Here the matrix $\theta$ is defined by $\theta_{ij}$ across high-dimensional vectors from 1 to $N$ assuming $x_i, x_j \in \mathcal{X}$ can be assigned labels (i.e. they are samples not features). UMAP allows for a "minimum distance" parameter that determines the minimum distance of points in low-dimensional space. Because Haisu reduces $\hat{p}_{ij}$ relative to $p_{ij}$, at higher values of $str$, one may want to increase "minimum distance" if a visualization similar to the unmodified embedding or preservation of global structure is desired.

UMAP is topologically motivated in that the method aims to identify a simplicial set covering the dataset that recovers the structure in the high-dimensional space and seeks to reproduce it in the projection. Haisu will affect the determination of the final embedding by determining the degree to which the topological structure of the manifold is preserved via the influence of the input hierarchy. Higher values of $str$, edge weights, and label probabilities for pairs $x_i, x_j$ will increase the propensity of similar classes in the hierarchy to exist as $k$ nearest neighbors in the manifold. This can affect the local-connectivity constraint $\theta_{ij} \cdot n_i$, and directly affects the directed edges in the weighted k-neighbor graph via $\hat{s}_{j|i}$ and the resulting probability that at least one of the two edges from $x_i$ to $x_j$ or $x_j$ to $x_i$ exists via $\hat{p}_{ij}$. Thus when using HAISU, UMAP is modified in a manner that the topological representation of the high-dimensional data and resulting embedding reflects the input hierarchy.

PHATE uses an $\alpha$-decaying kernel function with an adaptive bandwidth $\sigma_i$ (for a gaussian kernel, $\sigma_i = 2$). To better preserve global relationships, PHATE computes a diffusion geometry. We can apply $\theta_{ij}$ to the distance function $d(x_i, x_j)$ as with t-SNE and UMAP.

$$s_{j|i} = \exp\left(-\left(\frac{d(x_i, x_j)}{\sigma_i}\right)^{\alpha}\right)$$

$$\hat{s}_{j|i} = (s_{j|i})^{\theta_{ij}} = \exp\left(-\theta_{ij} \cdot \left(\frac{d(x_i, x_j)}{\sigma_i}\right)^{\alpha}\right)$$

$$\hat{p}_{ij} = 1/2(\hat{s}_{j|i} + \hat{s}_{i|j})$$

PHATE integrates the graph-tools library [22] where $\alpha$ is applied to the precomputed distance matrix, thus, the default effect of Haisu on $s_{j|i}$ is as follows which more closely preserves the effect of the decay factor $\alpha$. As a byproduct, this results in a less drastic hierarchical distancing effect for higher values of $\alpha$.

$$\hat{s}_{j|i} = (s_{j|i})^{\theta_{ij}^{\alpha}} = \exp\left(-\left(\frac{\theta_{ij} \cdot d(x_i, x_j)}{\sigma_i}\right)^{\alpha}\right)$$

In addition, for PHATE there will be an increased pairwise similarity between $x_i$ and $x_j$ reducing $\hat{p}_{ij}$ relative to $p_{ij}$. The degree of Haisu's modification $H$ is even greater than in t-SNE due to the exponential decay factor. Thus, due to the closer high-dimensional distances, Haisu will benefit from a lower user-provided $knn$ parameter and lower decay factor to prevent potential crowding in the embedding.

PHATE models a diffusion process aiming denoise the data and represent global and local information, using multidimensional scaling (MDS) to project the information to lower-

dimensional space. Like t-SNE, PHATE does not make any topological guarantees but encodes local data using local similarities and global data via potential distance via diffusion. Local similarities between $x_i$ and $x_j$ using Euclidean distances and the Markov-normalized affinity matrix using the $\alpha$-decaying kernel function $\hat{s}_{j|i}$ are influenced so that datapoints with closer labels in the hierarchy will have greater local similarities and higher initial random walk probabilities as a result. Potential distances employ the local similarities to determine $t$-step walk probabilities, hence, in the normalization of local affinities Haisu will also affect the global determination of PHATE as it contributes to the M-divergence. In this manner, PHATE's sensitivity to small distances in the probability distribution corresponding to larger global structures is especially impacted by Haisu through the decay factor $\alpha$. At high values of *str*, PHATE will often produce an embedding with branches that resemble the input hierarchy to a greater extent than UMAP or t-SNE given *self-distance* is sufficiently small or zero.

In review, the pairwise similarities for each NLDR method are formulated in a manner that the graph-based pairwise factor $\theta_{ij}$ does not compromise a method's primary architecture. Haisu modifies the transformation from global distances to local similarities solely in the pairwise distance function $d(x_i, x_j)$. Because our method is subtractive, the structure of a given method's lower dimensional embedding is identical to its unadulterated configuration when our strength factor *str = 0*. Finally, Haisu does not interfere with pairwise similarities in the low-dimensional embedding when high dimensional distances are not referenced. This further ensures that the appearance and structure of a Haisu embedding closely match those of its respective NLDR approach at any dimension.

## Results

### Haisu incorporates an *a priori* hierarchy into non-linear dimensionality reduction

Haisu is a generalizable extension to nonlinear dimensionality reduction for visualization that incorporates an input hierarchy to influence a resulting embedding. We achieve this through a hierarchical distancing factor $\theta$ which directly modifies an existing pairwise distance function $d$ used in the calculation of asymmetric similarities for two points in high-dimensional space prior to embedding. Specifically, Haisu's modification is $\theta_{ij} \cdot d(x_i, x_j)$ for a high dimensional dataset $\mathcal{X} = \{x_1, x_2, \ldots, x_N\}$. A parameter $str \in [0,1)$ can be supplied to modify the "strength" of the hierarchy's effect on the distance function. When $str = 0$, the original distance function is preserved, and the resulting embedding is unmodified relative to the raw NLDR technique. At $> 0$ $d$ is multiplied by an interpolated factor determined by normalized path distances in the input hierarchy and the value of *str*.

Haisu can be immediately applied to any NLDR technique compatible with the precomputed "path-pairwise" distance matrix $H = (\theta_{ij})$, where $H$ is multiplied by the unmodified distance matrix $D = (d_{ij})$. We illustrate the effect of Haisu on PHATE, t-SNE, and UMAP at multiple strength factors, and include PCA (executed on $H$) to express more simply how Haisu modifies the pairwise distance matrix. Strength values of 1 are not usually informative due the inverse exponential nature of Haisu's distancing factor $\theta$. In the case of PHATE, this is especially evident due to the exponential effect of the decay factor $\alpha$. Thus, for the figures presented, we pass 0.999 as the maximum strength value instead of 1.

### Haisu can work effectively on a range of different datasets via multiple NLDR methods

We analyzed the performance of our technique on single-cell RNA-sequencing datasets of three different modalities. First, we include fresh peripheral blood mononuclear cells (PBMCs)

from Zheng *et al*. to differentiate between T cell subtypes based on function [23]. Second, we apply Haisu to a proximity-based hierarchy for embryonic cardiac cells [24]. Last, we include a hierarchy based on epithelial differentiation in healthy and ulcerative colitis patients [25]. Finally, to demonstrate the efficacy of our approach, we progressively remove TA 1 cell labels from the epithelial dataset. We denote *str* as the strength of Haisu's hierarchical distancing factor $\theta$.

We chose to target the high-dimensional single-cell dataset specified in Zheng *et al*. due to the ambiguity between T cell subclasses in the raw t-SNE plot (Fig 2). CD4+ and CD8+ T cells in Zheng *et al* [23] are highly interspersed making it difficult to discern relationships between the subclasses. Additionally, there is a known relationship between the given classes [26], which motivated the application of a hierarchy-based model. Our visualizations are produced from the filtered "Fresh 68k PBMCs (Donor A)" dataset provided by 10x Genomics [27]. Prior to analysis, UMIs (unique molecular identifiers) under a target variance were removed (we chose 0.1) and then min-max normalized by cell. The graph that we specify for this dataset is derived from a gene-based hierarchy with explicit classes from Zheng *et al* and implicit parent classes such as "Lymphoid" and "Conventional T Cell" (Fig 2). We uphold parent-child relationships in the graph based on lineage and a dense set of nodes to distinguish between T cell types based on function. Through a purely lineage-based method, CD34+ should be in a different connected component, but we chose to keep it linked to the head node of the tree due to its early role in hematopoiesis.

Across NLDR methods, Haisu effectively separates different populations of T cells. For the PBMC dataset, lower strength values are more effective due to the lower inter-class variability of the processed UMI counts within the presence of outliers that decrease the size of the primary area of interest. By modulating the strength factor, we observe multiple effects on the data across DR methods. Firstly, original clusters of Monocyte, B, and Dendritic cells maintain relative distance and shape. Even when clusters of Monocytes migrate, local dendritic embedding remains consistent for strength values roughly less than 0.5. We also observe (*A*) that Cytotoxic T, NK, and Naïve Cytotoxic T cells remain closely intermingled until higher strength values where NK cells cluster nearby due to the hierarchical distancing factor. Finally, a group of Cytotoxic T cells (*B*) can be seen distinctively groups with neighboring Naïve Cytotoxic cells in PHATE (*str* = 0.4), UMAP (*str* = 0.6), and t-SNE (*str* = 0.25). This suggests that some Cytotoxic T cells are more like those labeled "Naïve Cytotoxic" than their own class and thus possibly mislabeled. Haisu in this effect can be used to encourage the re-evaluation of the prediction model for a given class. The Naïve T cell population (*C*) which is difficult to observe in raw t-SNE and UMAP clearly borders Naïve Cytotoxic and T Regulatory cells at *str* = 0.2 and *str = 0.25* respectively. This global relationship (*C*) breaks down at higher strength values due to the stronger influence of sibling nodes (see UMAP at *str* = 0.6 and t-SNE at *str* = 0.25), highlighting the importance of choosing a correct lower-influential strength value to discern subtle relationships with Haisu. Lastly, the hierarchical t-SNE (H-tSNE) implies two or more subpopulations of CD4+ Naive and CD4+ Memory T cells at *str* = 0.2 (*D*).

We include the embryonic cardiac cell dataset in Fig 3 for its inclusion of a proximity-based hierarchy that includes labels that correspond to the tissue region from which a cell sample was extracted [24]. As opposed to a largely inferred hierarchy influenced by gene expression such as in (Fig 2), the hierarchy of this dataset is influenced by real-world, physical distance in the heart sample. Because we do not include the unsupervised classes but rather the anatomical site hierarchy from Li *et al*, the graph-label assignment is truer to this mode of Haisu where label likelihood is not given. Thus, we suggest that our method is particularly valuable in this case where a gene-based embedding can be influenced by the relative distance between cell

extraction sites. We include all samples with the tissue tag listed in the graph, drop genes under a variance of 0.5, and perform min-max normalization per cell.

For this dataset, there are two primary observable clusters in each raw NLDR method. This separation is likely attributed to the two batches of e10.5 cells collected several months apart [24]. Haisu is influenced by this distancing at lower strength values in t-SNE and PHATE and maintains intra-cluster separation even up to $str = 0.999$. By introducing the hierarchical distancing factor, the ventricle subgroups are more discernable in both clusters (*A*). With NLDR techniques that maintain global structure more effectively such as PHATE and UMAP, Haisu is more informative between hierarchically distanced clusters. PHATE at $str = 0.5$ branches into three groups with two sub-groups for each same-label cluster. In this manner, Haisu rearranges the original graph based on the hierarchy but maintains the original global structure. For example, the atrioventricular canal cells are in each of the two primary clusters for the raw graph (*B*) and in PHATE at $str = 0.5$, they are in a branch but maintain relative distance (*B'*). UMAP at $str = 0.4$ demonstrates a similar effect (*B''*) but without the branching common to PHATE. For t-SNE, cells with siblings in the graph can easily become dramatically separated (*C*), at which point distances between the resulting clusters are less informative. Thus, we encourage lower strength values (ex $str = 0.3$) for t-SNE compared to PHATE and UMAP (*C'*).

Finally, in Fig 4 we investigate the application of Haisu to an epithelial differentiation dataset for cells during ulcerative colitis [25]. We filtered out genes under a variance of 1 and performed min-max normalization across cells on the resulting matrix. Without our method, the raw NLDR embeddings are too heterogeneous to discern any patterns in the data. Thus, we integrate the known hierarchy to create a more readable embedding. Most notably, by applying the hierarchy, we can observe sub-clusters that are not identifiable in the raw embeddings. Examining PHATE and UMAP at $str = 0.8$, we observe two primary groups of cells caused by two primary parent nodes in the graph: secretory TA cells and enterocyte progenitors. In PHATE at $str = 0.8$ and UMAP at $str = 0.7$ these two groups are split by the Stem and TA 2 labels (*A*), directly mirroring the structure input hierarchy graph. Just as with the cardiac dataset, we expect DR techniques that preserve global structure to better reflect properties of the input graph. Prior to the clumping of graph node siblings at $str = 0.9$, PHATE and UMAP allow for the inspection of more subtle inter-cluster relationships such as the position of Best4 + Enterocytes (*B*) among TA 1, M-like cells, enterocyte progenitors, and immature enterocytes 2 in PHATE $str = 0.8$. This suggests possible subpopulations or possible alternatives for the labeling of specific Best4+ Enterocytes that gravitate more toward the aforementioned classes. Smaller populations of cells such as M-like cells are also easier to view as the strength factor is increased (*C*). UMAP at $str = 0.8$ reveals a group of immature enterocytes 2 that are surrounded by enterocyte progenitor cells (*D*). This relationship is too loose to observe in the raw UMAP embedding at $str = 0$. As siblings in the hierarchy graph, these two labels stay close even at higher strength value up to $str = 0.9$. We similarly observe this same relationship in t-SNE at $str = 0.7$ as immature enterocytes are spread among smaller enterocyte progenitor clusters (*D'*). Here one can effectively employ the tendency of t-SNE to clump local groups for this dataset, finding clusters of cells with more than one class of cell, suggesting that the two (or more) classes share similar traits.

## Haisu supports semi-supervised analysis if hierarchical labels are only partially available

To empirically test the efficacy of Haisu's ability to preserve the structure and relationships between the modified embedding and the original, we illustrate the effect of removing a class label. In Fig 5, We progressively replace the label of TA 1 cells with a 'dummy' label that exists

in a disconnected component in the input hierarchy. In this manner, cells with the original label "TA 1" are completely removed from the influence of our hierarchical distancing factor and the original pairwise distance for that node is maintained. 100% removal indicates that all TA 1 cells in high-dimensional space are given the 'dummy' label. We suggest that even in an embedding highly influenced by the input graph (i.e., a higher value of *str*) Haisu maintains the relative positioning of datapoints even when they are unlabeled. This is a critical observation to communicate the efficacy of our approach such that Haisu at low strength values is a non-destructive method to influence an embedding based on a hierarchy. To alleviate intra-class clustering of points with the "Dummy" label, we specify Haisu construction parameter *self-distance* as 1 and include "Dummy" in the set of labels *avoid-self* as described in *Design and Implementation*. We also set *disconnected-distance* to 1 which encourages graph-disconnected TA 1 cells in the embedding to instead cluster near their next most similar class. In combination, these parameters ensure $\theta_{ij} = 1$ when the labels of $x_i$ or $x_j$ are "Dummy" which maintains high-dimensional distances from the raw NLDR approach.

In Fig 5 TA 1 cells are colored a darker gray when their label is provided to HAISU, and a lighter gray when it is discarded and given the label 'dummy.' To aid in identification of TA 1 labels and Cycling TA labels, we provide isolated label overlays for each graph in blue and red. Observing the relative position of TA 1 cells in the raw graphs of each NLDR method, we focus on a strong coupling with TA 2 Cycling TA cells (cluster A) closely distanced from the enterocyte progenitor cells as indicated by an arrow from cluster B. The TA 2 cell label is the grandparent of TA 1, so local clustering in t-SNE and global closeness in PHATE and UMAP is expected. This same relationship is expected to hold with enterocyte progenitors with are rewarded in Haisu because their cell label is the parent of 'TA 1.' However, Haisu does not encourage close positioning of TA 1 cells with Cycling TA cells in the embedding due to their distance in the graph, instead the raw high-dimensional distances are used. Thus, it is a remarkable testament to the preservation of global structure when we observe clustering of unlabeled TA 1 cells alongside Cycling TA at high strength values ($A'$, $B'$). For t-SNE and PHATE at 0% removal clusters $A'$ and $B'$ do not exist; rather cycling TA and TA 1 are completely separated and in their own clusters. However, once the TA 1 label has been disconnected in the hierarchy graph, we see these clusters form again, reflecting the structure of the original graphs while still maintaining the influence of the hierarchy for other class labels. Thus, at 100% removal we observe that the unlabeled TA 1 cells remain close in the embedding to enterocyte progenitors across strength values and NLDR techniques. For PHATE, cycling TA and TA 1 labeled points in the original embedding largely overlap (cluster C). As before, at 0% removal and $str = 0.9$, PHATE distances nodes based on the hierarchy and at 100% removal, cluster C is reestablished ($C'$).

Across each NLDR method, the impact of the hierarchical distancing factor at different strengths is readily apparent in the progression of randomized embeddings. We tend to favor a lower strength factor to "push" clusters away from distant nodes in the hierarchy while maintaining the nuances of the raw NLDR methods. In this regard, the influence of information not present in the hierarchy is still expressed in the higher dimensional manifold and visible within the embedding. The input graph and strength factor encourage a user-centric approach that allows the input of prior knowledge. Thus, one can manually adjust the strength factor to set the degree of separation when targeting specific relationships in the graph.

This approach can similarly be performed on each dataset for a gene-based hierarchy where sample classes are inferred with a given likelihood. Haisu in this mode can implement a probability-dependent approach that weights poor classifications less heavily. We first replace the ground-truth label of a sample $\ell_k$ with an inferred label $\ell'_k$ based on a given predictor. For example, we can predict by a cell's expression of key genes per labels. Hierarchy nodes in this

case could also be derived from representative proteins of the provided classes. Relevant UMI counts would first be summed for each node $v{\in}V$ of the gene-based graph following by the summation of key genes $g{\in}v$. This simple classifier would finally label the given sample based on the node with the highest normalized expression.

$$\ell'_k = \max(\sum_{v \in V} \sum_{g \in v} M_{gk})$$

Scores for each inferred class can then be normalized and influence the likelihood-based pairwise distance matrix described in the *Design and Implementation* section. Because the graph is also weighted with a strength factor, we can control this effect by modulating *str*. Inferred labels with low UMI counts contribute less to the hierarchical pairwise distance matrix, ensuring that the original embedding is upheld. Thus, the original structure can be preserved when the confidence of a given prediction is low. This prevents faulty clustering from the influence of the hierarchical pairwise distance matrix. With 100% confidence for inferred labels, results of the gene-based method would mirror those of the class-based hierarchy (Figs 2–4).

## Availability and future directions

Haisu formulates a direct relationship between the distance of two graph nodes in the hierarchy and the resulting pairwise distance in high-dimensional space ($\theta{\cdot}d$ respectively). We chose this approach to ensure that closely related sibling nodes are encouraged in the embedding and farther nodes produce a noticeable amount of spreading. However, the effect of the hierarchy can easily be modified for a desired visualization. To better express a hierarchy in an embedding, one could introduce a diminishing factor such that nodes deeper into the graph have a reduced influence as opposed to the base classes. An application of such a strategy is relevant with the presence of distinct classes containing many nuanced subclasses as children. Another simple modification would be the application of a weighted graph in the calculation of the shortest path distance $\overrightarrow{j'i'}$ between labels $i'$ and $j'$.

As described in the *Design and Implementation* section, our distancing algorithm is subtractive in nature. Rather than modifying the distance factor by an arbitrary amount, we perform a linear interpolation from 0 to 1 based on the relative graph node distances. At zero, the original pairwise distances are upheld, and at one, pairwise distances are multiplied by normalized graph distance. We suggest that this strategy is more effective in maintaining the local and global structure of the data at all choices of strength factor. The crux of Haisu's practical use is, thus, in the choice of an appropriate strength factor. Primarily, one should employ this variable to control the extent of hierarchical distancing desired. We found its effect to be highly dependent on the characteristics of the dataset and the size/structure of the input hierarchy. In the included figures, we highlight lower strength factors that result in an embedding with similar visual structure to the base DR technique. Empirically, we suggest choosing a strength prior to significant clumping of neighboring nodes in the hierarchy graph. To aid in identification a desired cluster overlap, we include a visual analysis that wraps clusters for each label with a convex hull and computes the nearest shape intersection (Fig C in S1 Text). In Supplementary Note 1 in S1 Text we also describe how a strength value can be determined with respect to a hierarchically modified silhouette metric.

We chose to modify the pairwise distance function prior to embedding due to the ease of generality into multiple NLDR methods. Additionally, due to possible random initialization that follows the initial pairwise distance calculation in most methods, label ordering can quickly become impossible to assign. However, with further modification of the base NLDR

algorithms, one could instead modify the distance function, influence the initial random embedding, or extend the objective function with a hierarchical factor. We also chose not to integrate directed graphs with this approach to support more complicated inter-class relations, but one could extend the approach to punish a pairwise-comparison when moving against the directed graph. This would result in more inter-class separation and less intra-class separation which could be valuable for some datasets.

Computationally, Haisu mirrors the limitations of the integrated NLDR approach spatially and temporally. To mitigate runtime considerations, we minimize function calls, cache hierarchy node distances, and intern string dictionary lookups. Furthermore we parallelize computation of the shortest-path pairwise triangle matrix and provide access to the 'mpi' variable of the scipy.spatial.distance function [21]. As with t-SNE, UMAP, and PHATE, appropriate memory must be available to compute and process a distance matrix for large datasets. Haisu introduces additional overhead for the hierarchical distancing matrix $\theta$ equal to $16 \times \ell^2$ bits where $\ell$ is the total number of labels provided. We have included Haisu's distance matrix calculations in Table A in S1 Text for the datasets processed in the Results section to demonstrate run-time performance.

In this study, we primarily investigate the use of Haisu as a cross-domain visualization tool. However, future work could integrate Haisu into a semi-supervised classification scheme. suggest that for datasets in domains outside of scRNA-sequencing where labels assignments can be given with 100% certainty, Haisu could be effective even at higher dimensions for clustering. We suggest that clustering and processing at higher dimensions is feasible and could be an effective, supervised intermediate step because Haisu preserves the dimensionality reduction-specific architecture.

We proposed a novel general-purpose approach for nonlinear dimensionality reduction that incorporates an input hierarchy. We modify the premier visualization-centric techniques in this field and demonstrate our results on real-world single-cell datasets. Furthermore, we introduce a feature-based modification that enables users to integrate our method with weighted class labels.

## Supporting information

**S1 Text. Supplementary information. Table A in S1 Text:** Mean runtimes (mm:ss.s) correspond to Fig 2,3,4 for the Haisu hierarchy-base pairwise distance matrix and downstream choice of NLDR method. Each method was run with no subsampling across 48 cores and respective parallel program parameters. T-SNE, UMAP, PHATE, and PCA runtimes do not include the pair-wise distance calculation time and (other than PCA) are run with the 'precomputed' metric parameter*. Thus, the runtime of Haisu is contingent on the size of the input dataset and the accompanying metric used. We have provided parallel and optimized single-thread versions to facilitate processing large datasets. **Table B in S1 Text: Haisu better represents a hierarchy graph in an embedding when compared to equal-weighted, independent classes.** We take the area under each graph (AUC) integration using Simpson's rule. Scores are averaged for strength factors from 0 to 0.999 or integrated by a step size of 0.1. We include HS* = 1-HS such that higher scores (bolded) are more favorable, indicating that Haisu more accurately represents the hierarchy graph while maintaining integrity of the embedding. **Table C in S1 Text: Haisu better represents a hierarchy graph in an embedding when compared to equal-weighted, independent classes.** We compare methods by nearest neighbor distribution similarity with the input hierarchy graph (HC) and by a hierarchically modified silhouette score (HS). For every dataset and scoring metric, Haisu demonstrates significant benefits for representing the input hierarchy as compared with equally weighted classes. HC*

$= 1-\sqrt{(HC)}$ and HS* = 1-HS such that higher values of HC* and HS* are favorable. **Table D in S1 Text: No NLDR method is consistently more compatible with Haisu.** We compare methods by nearest neighbor distribution similarity with the input hierarchy graph (HC) and by a silhouette score (HS). HC is calculated across KNN = 3, n/100, n/10 where n is the total number of samples for a given dataset and KNN is the k-nearest neighbors including self. Graphs are calculated for strength factors from 0 to 0.999 by a step size of 0.1. We take the area under each graph (AUC) integration using Simpson's rule. $HC^* = 1-\sqrt{(HC)}$ and HS* = 1-HS such that higher values of HC* and HS* are favorable. **Fig A in S1 Text:** We display the full progression of Haisu from a strength factor of str = 0 to str = 0.999 by a step size of 0.1 corresponding to Fig 2,3,4. Modified PCA, PHATE, t-SNE, and UMAP are displayed from top to bottom for each dataset. **Fig B in S1 Text:** Full progression of Haisu strength factor values and percentage removal of the TA-1 label corresponding to Fig 5. Modified PHATE, t-SNE, PCA, and UMAP nonlinear dimensionality reduction techniques are depicted. The y-axis indicates str value and the x-axis indicate percentage removal of the TA 1 label. **Fig C in S1 Text**: We demonstrate a heuristic for aiding in the choice of hierarchical distancing factor (str). For the epithelial dataset (shown), we favor overlap scores between 35% and 45%, which visually demonstrate hierarchically influenced cluster separation while maintaining characteristics of the raw embedding. To find mean overlap, we compute the nearest shape intersection for each cluster's convex hull. Clusters are determined by the input classes and outliers are first removed by z-score. **Fig D in S1 Text: The h-silhouette score can be used to find an ideal range of Haisu strength values.** We demonstrate a hierarchically modified silhouette score (h-silhouette) that determines how effective a given strength value is at maintaining inter-class distance in the embedding, weighted by the input hierarchy. A score closer to zero is more desirable. **Fig E in S1 Text: Haisu better represents the expected hierarchy labels in local and global neighborhoods when compared to equal-weighted, independent classes.** We calculate the modified "HC" score for each dataset across multiple choices of k-nearest neighbors (KNN). HC is determined by calculating the mean chi-squared statistic for the observed v.s. expected distribution of k labels surrounding each point in the embedding. We report 1 minus the square root of the resulting score (y-axis) s.t. values closer to 1 are more desirable. The expected distribution reflects the input hierarchy graph l-1 normalized from 1/maxdist to 1. HC is calculated across KNN = 3, n/100, n/10 where n is the total number of samples for a given dataset and KNN is the k-nearest neighbors including self.
(DOCX)

## Acknowledgments

We thank Didem Çobanoğlu for her extensive guidance in the formulation of this approach and relevant biological expertise. We also thank Venkat Malladi and Spencer Barnes for assistance in selecting the single-cell RNA sequencing datasets and visualization methods.

## Author Contributions

**Conceptualization:** Kevin Christopher VanHorn, Murat Can Çobanoğlu.

**Data curation:** Kevin Christopher VanHorn, Murat Can Çobanoğlu.

**Formal analysis:** Kevin Christopher VanHorn.

**Funding acquisition:** Murat Can Çobanoğlu.

**Investigation:** Kevin Christopher VanHorn.

**Methodology:** Kevin Christopher VanHorn.

**Project administration:** Murat Can Çobanoğlu.

**Resources:** Kevin Christopher VanHorn.

**Software:** Kevin Christopher VanHorn.

**Supervision:** Murat Can Çobanoğlu.

**Validation:** Kevin Christopher VanHorn.

**Visualization:** Kevin Christopher VanHorn.

**Writing – original draft:** Kevin Christopher VanHorn.

**Writing – review & editing:** Kevin Christopher VanHorn, Murat Can Çobanoğlu.

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
