## [Decision Letter · Decision Letter 0]

6 Apr 2022

Dear Dr Cobanoglu,

Thank you very much for submitting your manuscript "Haisu: Hierarchically Supervised Nonlinear Dimensionality Reduction" for consideration at PLOS Computational Biology.

As with all papers reviewed by the journal, your manuscript was reviewed by members of the editorial board and by several independent reviewers. In light of the reviews (below this email), we would like to invite the resubmission of a significantly-revised version that takes into account the reviewers' comments.

We cannot make any decision about publication until we have seen the revised manuscript and your response to the reviewers' comments. Your revised manuscript is also likely to be sent to reviewers for further evaluation.

Sincerely,

Dina Schneidman

Software Editor

PLOS Computational Biology

Reviewer's Responses to Questions

**Comments to the Authors:**

Reviewer #1: The authors proposed an interesting approach for conducting nonlinear dimensionality reduction by incorporating hierarchical label and tested its performance on three single-cell RNA-sequencing datasets (embedded with commonly-used packages like t-SNE, UMAP, and PHATE). The concept of the proposed technique is generally straightforward and well described like investigating the embedding effect of TA1 cells without labels, explaining how to determine hierarchical distancing factor(str), and the main findings are validated with multiple strategies. The authors modify the premier visualization-centric techniques in this field, which has a significant merit and may potentially become a popular tool, if fully validated, for handling an overwhelming and rapidly-increasing amount of single-cell sequencing data. However, I do have some concerns before considering publication:

1. If the hierarchical label information is inserted into the step of “feature selection”, will the pipeline generate same or similar results? Moreover, if the user-defined (a prior called by the authors in the manuscript) hierarchical information is applied for dimensionality reduction, would it lead some leakage of labeling info for subsequent unsupervised clustering?

2. One of major advantages by performing single-cell sequencing is to break through the current hierarchical understanding of cell differentiation. Here it is applied as a prior knowledge, would it become an obstacle to uncovering new insights?

3. It seems that the selection criteria of hierarchy relationships in three datasets are different, will this treatment cause some biased outcomes？And, is there any principled guidance to help determine the most appropriate hierarchy labels for independent datasets？Line 339 on Page 16:“However, the effect of the hierarchy can easily be modified for a desired effect.”How to help users to walk around this trap?

4. Line 354 on Page 17: “We found its effect to be highly dependent on the characteristics of the dataset and the size/structure of the input hierarchy.” I wonder what aspect of the characteristics of a new dataset will affect the size/structure of the input hierarchy and how?

5. Regarding the definition of the distance modification function in Results (Line 109-112 on Page 7), it is given as follows:

Is it in effect equivalent to the equation (Line 421-424 on Page 18)

Reviewer #2: Real world data often lies in low-dimensional manifolds embedded in high-dimensional spaces, with high codimensions (manifold hypothesis). It is generally a hard task to obtain projections from high-dimensional spaces to visualizable spaces, e.g. of dimension 2 or 3, such that the properties that characterize the low-dimensional embedding are preserved. There are several techniques employed for dimensionality reduction, typically divided into linear and nonlinear classes. The latter class includes t-SNE, UMAP and PHASE.

In the article under consideration, the authors propose a method that modifies the distances used in t-SNE, UMAP and PHASE by means of user-defined hierarchies. This method results in a modification of the projection that reduces the dimensionality, parametrized by a strength factor decided by the user. The main focus of this approach is on biomedical data, and the authors provide experiments on single-cell RNA sequencing datasets.

The article is of value, and it should be considered for publication in the journal PLOS Computational Biology. However, the article currently needs significant revisions, as described below. The main issue lies in the fact that the method is overall not well explained, and some natural questions regarding the relation between t-SNE, UMAP, PHASE and the corresponding modified versions proposed in this article are not properly discussed. Also, the organization of the article does not seem to be ideal, and it would be greatly beneficial to restructure it.

-- The subsection named “Hierarchically Supervised NLDR” is not working at the moment. There are several issues with the exposition. For instance, the setting is not well explained, and it can only be guessed by the reader. There is a high dimensional input dataset, but it seems that the input should also consist of a hirarchy graph. This is mentioned at the end of the first paragraph (line 100), but it should be manifest from the beginning. It is not clear what “variable distance function” means here. It seems that variable here means “not constant”. However, distances cannot be constant, as this would violate the triangular axiom. So, one might deduce that variable here means something else, and the authors are trying to convey that the distance satisfies this property of being variable. Also, the term “distance modification” appearing in line 98 is also unclear at this point. It is clarified (implicitly) towards the end of the manuscript, in the Methods section, but it should not be left unclear for so long. The graph “G” seems to be the hirarchy graph mentioned above. This should be explicitly stated.

Similarly, the term ``strength’' in line 106 does not correspond to any canonical notion in the computational sciences. The meaning can be deduced afterwards from line 110, but again, this is not helping the reader. More generally, in this first subsection, the relation between theta_{ij} and the methodology is completely unclear. It is only understood that it modifies the distance, but it is not clear how until the last section of the article. This subsection could be merged with the last section for clarity, and rewritten. If the authors prefer to discuss the technicalities after showing the results, this could be done too. A brief general description without using equations could be added as the first subsection of the article, and a detailed account (current first subsection and last section merged together and better exposed) could appear at the end. The current situation lies somehow in between.

In line 115, the use of s.t. (such that), does not seem to follow the traditional practice in mathematics. It seems rather that the authors meant to say “so that”. But s.t. means “such that”, which is also indicated by a vertical bar in sentences using the formalism of mathematical logic. In line 121, it seems that the quantity “m” could be obtained via min, max or other types of aggregation. The authors say that the probability could be taken to be min, max or other types of aggregation. This does not seem to be correct. Also, “m” seems to depend on i,j, so this dependence should be explicitly shown in the notation because “m” is not a parameter common to all x_i, x_j.

-- The Benchmark subsection is overall well written. However, the tests that have been to evaluate the ability of this method to preserve the structure of the embeddings generated by t-SNE, UMAP, PHASE do not seem to be complete. The results regarding the relative positioning of points found here (line 269 on) are encouraging, but the aforementioned methods were motivated by deep analytical, geometrical and topological reasons which are not considered here. It would be reasonable to expect that increasing the strength coefficient the perturbation on the distances increasingly alters the geometric structure until the latter is lost. If on the one hand it would be unreasonable to expect a full understanding of these mechanisms in this article, it is on the other hand good to provide at least a perspective on this matter, as it can be an important factor in the fine tuning of the parameters for specific applications.

-- There is an issue with considering the ``Effect of HAISU on t-SNE, UMAP, and PHATE’’, as in the last section of the article. Until now, no general understanding of the method has been provided to the reader. So, as far as the reader knows, up to now, HAISU is a method that alters t-SNE, UMAP, and PHATE to produce better embeddings. In fact, the only explanation regarding the way the coefficients theta_{ij} descibed at the beginning of the article influence the projection is provided in this section. In other words, this is the actual definition of the method, not the analysis of the effect of the method on t-SNE, UMAP, and PHATE. The method is not considered in generality, even though some experiments on PCA are also provided to compare to linear methods. One could only guess how the definition goes in the case of PCA. From the codes one might get a clearer understanding of how to apply HAISU in general, but it is not ideal.

-- The right hand sides of the equations in lines 419 and 422 seem identical. It seems that there is a factor of theta_{ij} in line 419 that should not appear.

**Have the authors made all data and (if applicable) computational code underlying the findings in their manuscript fully available?**

Reviewer #1: Yes

Reviewer #2: **No: **

PLOS authors have the option to publish the peer review history of their article (what does this mean?). If published, this will include your full peer review and any attached files.

Reviewer #1: **Yes: **Zheng Wang

Reviewer #2: No
---

## [Decision Letter · Decision Letter 1]

3 Jul 2022

Dear Dr Cobanoglu,

We are pleased to inform you that your manuscript 'Haisu: Hierarchically Supervised Nonlinear Dimensionality Reduction' has been provisionally accepted for publication in PLOS Computational Biology.

Best regards,

Dina Schneidman

Software Editor

PLOS Computational Biology

Reviewer's Responses to Questions

**Comments to the Authors:**

Reviewer #1: The authors have addressed my concerns.

**Have the authors made all data and (if applicable) computational code underlying the findings in their manuscript fully available?**

Reviewer #1: Yes

PLOS authors have the option to publish the peer review history of their article (what does this mean?). If published, this will include your full peer review and any attached files.

Reviewer #1: **Yes: **Zheng Wang

---

## [Editor Report · Acceptance letter]

15 Jul 2022

PCOMPBIOL-D-21-02208R1 

Haisu: Hierarchically Supervised Nonlinear Dimensionality Reduction

Dear Dr Cobanoglu,

I am pleased to inform you that your manuscript has been formally accepted for publication in PLOS Computational Biology. Your manuscript is now with our production department and you will be notified of the publication date in due course.

With kind regards,

Zsofia Freund
